# Experimental Evolution of Magnetite Nanoparticle Resistance in *Escherichia coli*

**DOI:** 10.3390/nano11030790

**Published:** 2021-03-19

**Authors:** Akamu J. Ewunkem, LaShunta Rodgers, Daisha Campbell, Constance Staley, Kiran Subedi, Sada Boyd, Joseph L. Graves

**Affiliations:** 1Department of Nanoscience, University of North Carolina at Greensboro, Greensboro, NC 27401, USA; judeakamu@gmail.com; 2Department of Biology, University of North Carolina at Greensboro, Greensboro, NC 27412, USA; Lmrodger2@gmail.com; 3Department of Chemical, Biological, and Bioengineering, North Carolina A&T State University, Greensboro, NC 27411, USA; dmcampb2@gmail.com; 4Department of Chemistry, Bennett College, Greensboro, NC 27401, USA; cbstaley26@gmail.com; 5College of Agricultural and Environmental Sciences (CAES), North Carolina A&T State University, Greensboro, NC 27411, USA; ksubedi@ncat.edu; 6Department of Ecology and Evolutionary Biology, University of California, Los Angeles, CA 90095, USA; smboyd1@g.ucla.edu; 7Department of Biology, North Carolina A&T State University, Greensboro, NC 27411, USA

**Keywords:** *Escherichia coli*, magnetite nanoparticles, metals, antibiotics, genomics, pleiotropy, cell morphology

## Abstract

Both ionic and nanoparticle iron have been proposed as materials to control multidrug-resistant (MDR) bacteria. However, the potential bacteria to evolve resistance to nanoparticle bacteria remains unexplored. To this end, experimental evolution was utilized to produce five magnetite nanoparticle-resistant (FeNP_1–5_) populations of *Escherichia coli*. The control populations were not exposed to magnetite nanoparticles. The 24-h growth of these replicates was evaluated in the presence of increasing concentrations magnetite NPs as well as other ionic metals (gallium III, iron II, iron III, and silver I) and antibiotics (ampicillin, chloramphenicol, rifampicin, sulfanilamide, and tetracycline). Scanning electron microscopy was utilized to determine cell size and shape in response to magnetite nanoparticle selection. Whole genome sequencing was carried out to determine if any genomic changes resulted from magnetite nanoparticle resistance. After 25 days of selection, magnetite resistance was evident in the FeNP treatment. The FeNP populations also showed a highly significantly (*p* < 0.0001) greater 24-h growth as measured by optical density in metals (Fe (II), Fe (III), Ga (III), Ag, and Cu II) as well as antibiotics (ampicillin, chloramphenicol, rifampicin, sulfanilamide, and tetracycline). The FeNP-resistant populations also showed a significantly greater cell length compared to controls (*p* < 0.001). Genomic analysis of FeNP identified both polymorphisms and hard selective sweeps in the RNA polymerase genes *rpoA*, *rpoB*, and *rpoC*. Collectively, our results show that *E. coli* can rapidly evolve resistance to magnetite nanoparticles and that this result is correlated resistances to other metals and antibiotics. There were also changes in cell morphology resulting from adaptation to magnetite NPs. Thus, the various applications of magnetite nanoparticles could result in unanticipated changes in resistance to both metal and antibiotics.

## 1. Introduction

Both ionic and nanoparticle iron have been proposed as materials to control multidrug-resistant (MDR) bacteria [1,2,3]. This idea may seem contradictory in that all organisms require iron as an essential micronutrient. It often serves as an enzymatic co-factor, and therefore its intracellular concentration must be tightly regulated in order to maintain cell viability [4]. Iron is one of the most important micronutrients as it fulfills many biological roles [5]. Iron-containing proteins (heme proteins, iron-sulfur cluster proteins, and di-iron and mononuclear enzymes) play roles in nitrogen fixation and metabolism, and serve as electron carriers for respiration [6]. However, despite its critical role in bacterial metabolism, acquiring iron is the greatest challenge for bacterial growth [4,7]. Therefore, iron deficiency is one of the most common nutritional stressors, especially in aquatic environments [8]. The fundamental requirement for iron plays a crucial role in microbial pathogenesis, thus many organisms utilize iron sequestration as a defense against infection [9,10].

Yet, despite iron being an essential metal, it can also be extremely toxic under aerobic conditions [11]. There are multiple mechanisms of excess iron toxicity (see Table 1). For this reason, iron homeostasis is tightly regulated. Such tight regulation associated with the bacterial requirement to acquire iron from its environment can potentially be used as an evolutionary trap. Evolutionary traps are defined as a situation in which an organism prefers resources that reduce its fitness [12]. Thus, in an environment where iron is present in excess, we can ask the question whether bacteria have the capacity to control either the uptake, storage, or efflux of iron at rates sufficient to avoid toxicity? Alternatively, it is important to ask: can and how do bacteria evolve resistance to excess iron? In addition, if they can, what correlated traits will result from the adaptation? We have already answered some of these questions in a series of experiments utilizing ionic iron II, iron III, and the iron analog gallium III [13,14,15]. These studies utilized experimental evolution and examined mechanisms of iron and gallium resistance through the evaluation of phenotypic and genomic changes in *E. coli* K-12 MG1655. We found that this strain of *E. coli* was capable of evolving resistance to excess iron and gallium primarily through changes in the uptake of iron (or gallium). In addition, we found that iron (II)- and iron (III)—resistant populations showed unanticipated correlated resistances to a range of antibiotics including ampicillin, chloramphenicol, rifampicin, sulfanilamide, and tetracycline. Our results suggested that *E. coli* K-12 could respond to excess iron by both physiological acclimation and evolutionary adaptation [13]. This was manifested by both patterns of genomic and gene expression changes exhibited in the iron (II)-, iron (III)-, and Gallium (III)-resistant populations compared to their ancestors and controls (grown in the absence of excess iron or gallium).

In this study, we examine whether experimental evolution against spherical magnetite nanoparticles (FeNPs) will result in similar outcomes as those observed resulting from selection for ionic iron (II, III) resistance. Nanoparticles (NPs) are particles between 1 and 100 nanometers in size [16]. NPs are broadly classified into various categories depending on their morphology, size, and chemical properties [17,18]. Metallic ferrous nanoparticles are one of the most studied nanomaterials against multidrug-resistant bacteria [19]. The three most common naturally occurring forms of metallic ferrous NPs are hematite (α-Fe_2_O_3_), maghemite (γ-Fe_2_O_3_), and magnetite (Fe_3_O_4_ or FeO·Fe_2_O_3_). Magnetite is a black iron oxide, commonly called Hercules stone, and possesses the strongest magnetic behavior [20]. Magnetite is considered as a charge frustrated iron oxide due to the distribution of both iron (II) and (III) in crystallographic sublattice sites [21].

Magnetite NPs exist in nature, and engineered FeNPs are widely used in a range of applications due to their advanced optical properties [18]. FeNPs can be used to determine oxygen concentrations [22,23,24]. FeNPs are involved in orientation, navigation, and iron metabolism in prokaryotic and eukaryotic cells [22,25]. FeNPs are extensively used for applications in drug delivery [26], treatment of cancer cells [27], antimicrobials [28,29,30], and in general biomedicine and bioanalytics [31]. FeNPs are also now use for antimicrobial applications due to their very small in size and high surface area to volume [32,33]. FeNPs disseminate ionic iron II and iron III species with low toxicity to eukaryotic host cells minimizing undesirable side effects [29,34,35,36]. The antibacterial properties of the FeNPs have been tested against *Bacillus cereus*, *Escherichia coli*, *Staphylococcus aureus*, *Pseudomonas aeruginosa*, and *Serratia marcescens.* They have also been shown to effective against the yeast species *Candida albicans* [34,37].

However, none of these studies considered the possibility that bacteria could evolve resistance to FeNPs. Therefore, it is of paramount importance to describe mechanisms by which bacteria can become resistant to magnetite NPs and determine if resistance to magnetite nanoparticles confers resistance to ionic metals and antibiotics in order to fill gaps in our knowledge. In this study, we investigated how rapidly bacteria can evolve resistance to FeNPs and evaluated the nature of the genomic changes responsible for that resistance. In addition, we wanted to determine if magnetite resistance conferred correlated resistances to ionic metals and antibiotics. Finally, we wished to determine the similarity of phenotypic and genomic changed due to magnetite to those produced by ionic iron (II) and (III) resistance.

## 2. Materials and Methods

### 2.1. Bacterial Strains

*E. coli* MG1655 (ATCC #47076) was used in this study because it does not have plasmids and its circular chromosome is composed of 4,641,652 nucleotides (GenBank: GenBank: 117 NC_000913.3; Riley et al. 2006). All of our previous studies of ionic and nanoparticle resistance have used this strain [13,14,15]. This strain is the ancestor of all the selection treatments (FeNP and Controls) evaluated in this study.

### 2.2. Evolution Experiment

*E. coli* MG1655 (ATCC #47076) was routinely cultured in Davis Minimal Broth (DMB; Difco™ Sparks, MD, USA) with 1 g per liter of Dextrose (Dextrose, Fisher Scientific, Fair Lawn, NJ, USA) as the sole carbon source. The broth was supplemented with thiamine hydrochloride 10 μL in a final volume of 10 mL of total culture maintained in 50 mL Erlenmeyer flasks. The bacteria were incubated at 37 °C in broth at 150 rpm overnight and propagated by transferring 0.1 mL into 9.9 mL of fresh sterile DMB daily. We established 5 populations within the magnetite NP selection treatment. Each population was founded from a unique colony resulting from serial dilution after 24-h growth of the ancestral *E. coli* K-12 MG1655 sample. These replicates were exposed daily to 750 mg/L 20 nm spherical PVP-coated FeNPs (designated FeNP_1–5_). The colloidal magnetite nanoparticles (concentration of 20 mg/mL aqueous in 2 mM sodium citrate) were obtained from Nanocomposix (San Diego, CA, USA). The concentrations utilized to initiate the selection experiment were determined by minimum inhibitory assay (MIC, described below). The controls (designated C_1_–C_5_) were cultured in standard DMB medium without the addition of magnetite nanoparticles. 

### 2.3. Determination of Minimal Inhibitory Concentration (MIC) of Magnetite NPs

Minimum inhibitory concentration (MIC) values were assessed by measuring the growth rates of *E. coli* K-12 MG1655 by estimating their optical density at 625 nm (OD_625_). OD_625_ is an absorbance measurement at a wavelength of 625 nm in an accuSkan Go spectrophotometer (Fisher Scientific, USA). MIC is defined as the lowest concentration of an antimicrobial that will inhibit the visible growth by ~90% after 24 h of incubation at 37 °C compared with a growth control [38]. These values were determined by broth dilutions in a 96-well microtiter plate format consisting of ten concentrations (0–5000 mg/L) in triplicates of magnetite NPs. The plates were incubated at 37 °C and agitated at 600 rpm in an I2500 Series incubator shaker (New Brunswick Scientific, Enfield, CT, USA). The OD_625_ was measured after incubation at 37 °C for 24 h. The MIC was determined to be 1000.0 mg/L. A sub-MIC value (750 mg/L) was chosen to initiate the selection experiment. This value allowed for the initial growth of cultures without causing their extinction.

### 2.4. Phenotypic Assays: 24-h Growth

To determine if magnetite NP resistance confers correlated resistances to other metals and antibiotics, assays were conducted to assess 24-h growth to different concentrations of ionic forms of metals [Ag (I), Fe (II), Fe (III), Ga (III)] and the antibiotics [ampicillin, chloramphenicol, rifampicin, tetracycline]. These heavy metals and antibiotics are known to have different mechanisms of action. The ancestral strain was grown for 24 h in DMB broth and serial dilution was used to pick 5 independent colonies. These in turn were used to found replicates grown for 24 h and placed into the various phenotypic assays. The FeNP and control populations were sampled at 25 days in their various media for use in these assays. Bacterial growth was assessed by measuring at turbidity at 625 nm for hours 0 and 24 h, using a 98-well plate format accuSkan Go spectrophotometer (Fisher Scientific, USA) using clear polyester 98-well plates.

### 2.5. Genomic Analysis

Whole genome resequencing was utilized to identify genomic variants associated with FeNP and control populations. DNA was extracted from each replicate population at 25 days of selection in magnetite NP using EZNA Bacterial DNA kit (OMEGA) following the manufacturer’s instructions. DNA concentrations were determined fluorometrically using the QuantiFluor^®^ ONE dsDNA System (Promega Corporation, Madison, WI, USA) on a Quantus^®^ fluorometer (Promega Corporation). Measurements were performed according to the manufacturer’s recommendations. Genomic libraries were prepared using the standard protocol as described in the manufacturer’s instructions (Illumina, Nextera DNA Flex Library Prep Reference Guide). All the reagents used are included in the Nextera DNA Flex kit (Illumina, cat. Nos. 20,018,704, 20,018,705). 

The quality of the final library was verified using D1000 Screen Tape (Agilent Technologies, Santa Clara, CA, USA) following the manufacturer’s instructions. Library concentration was measured using the fluorometric quantification using the dsDNA binding dye as previously described and diluted 12pM with Resuspension Buffer. The pooled libraries were then run on an Illumina MiSeq (Illumina, San Diego, CA, USA) using the MiSeq v3 reagent kit. The depth of coverage of the sequencing runs ranged from ~20X to ~80X, with most exceeding 40X coverage. The SRA accession number for sequencing data is PRJNA694183 (FeNP-resistant, FeNP_1–5__D28 and controls, C_1–5__D28).

Sequence alignment and variant calling from the samples were achieved by use of the breseq 0.30.0 pipeline set to polymorphism mode (-p) and default parameters [39]. The pipeline makes use of three types of evidence to predict mutations, read alignments (RA), missing coverage (MC), and new junctions (JC) [40]. Reads that show a distinction between the sample and the reference genome that cannot be resolved to describe precise genetic changes are considered “unassigned” and would not be described nor interpreted. Finally, we report here only genomic variants that were not present in the ancestral population. Variants in our ancestral founding populations that differ from the *E. coli* K-12 MG1655 reference genome are reported in our previous studies [41,42].

### 2.6. Preparation of Cells for Scanning Election Microscope

Scanning electron microscopy (SEM) is widely used to measure changes in morphology of bacteria [43]. After resistance to FeNPs was observed (at 25d of selection) the outer morphology of the bacterial cells was examined using a Carl Zeiss Auriga-BU FIB FESEM (FESEM) (Carl Zeiss, Jona, Germany). Briefly, bacterial samples from each population were placed on cover slips after 10 min of incubation at room temperature, the samples were gently removed. Thereafter, the bacteria were fixed with Karnovsky fixative and incubated at 4 °C overnight. Following incubation, the samples were dehydrated with graded ethanol and then air-dried. The samples were then sputtered to avoid charging in the microscope. The images were acquired at a working distance of 7 mm and an accelerating voltage of 3 kV.

### 2.7. Inductively Coupled Plasma Optical Emission Spectrometry (ICP-OES) Analysis

The amount of ionic materials released into solution by metallic nanoparticles vary considerably depending upon conditions of the medium [44]. For this reason, an ICP-OES study was performed to determine or define the sublethal concentration of iron in samples utilized for MIC assays at 12 and 24 h. Operating conditions are shown in Appendix A
Table A1. Each sample was carefully pipetted and transferred to digestion tubes followed by the addition of concentrated nitric acid (67–70)% purchased from Fisher Scientific. After allowing for a 15 min pre-digestion procedure, the samples were digested in an automated sequential microwave digester, MARS 5 (CEM Microwave Technology Ltd., Matthews, NC, USA). The final product was a clean, transparent aqueous solution which was further diluted to a volume of 50 mL using double DI water. The concentration of acid in the final solution was <5%. A set of matrix matched standards were prepared to set up a calibration curve. All the samples were analyzed by using Optima 8300 (PerkinElmer, Inc. Shelton, CT, USA) in axial mode.

### 2.8. Statistical Analysis

Statistical analysis of the effect of selection regime and concentration (and their interaction) for all 24-h growth data was performed via General Linear Model utilizing SPSS version 26 (SPSS Inc., Armonk, NY, USA). All graphs in this paper were made via Prism 9 software. Finally, the phenotypic data from these studies will be submitted into DRYAD (https://datadryad.org/ accessed on 17 March 2021) upon acceptance of this manuscript for publication.

## 3. Results

### 3.1. Escherichia coli Can Rapidly Evolve Resistance to Stressful Levels of Magnetite NPs

The FeNP populations showed statistically significantly greater 24-h growth compared to the controls with increasing concentration in every substance tested. This was determined by using a general linear model that examined the effect of population, concentration of substance, and their interaction. The population effect examines whether the population a 24-h growth value was obtained from played a statistically significant role in determining its outcome. The concentration effect examines whether the concentration at which the growth value was obtained played a significant role in determining the outcome. The interaction effect examines whether both populations displayed the same response to concentration. The F and *p* values for all phenotypic assays (antibiotics and metals) are given in Table 2a. The controls showed statistically significantly greater growth compared to the ancestors for all substances tested except for ampicillin and rifampicin. As the comparison for magnetite, silver (I), and chloramphenicol was not as clear, general linear model results are reported for these substances in Table 2b. The 24-h growth after 25 days of selection in magnetite showed that FeNP populations exhibited a highly significantly superior growth compared to the control and ancestral populations across all concentration of magnetite NP (Figure 1). The FeNP populations show statistically significant greater growth than either the controls or ancestors from 250 to 1750 mg/L. The 24-h growth of the FeNP population increased across this range while that of the control and ancestors decreased until there was very little growth from 500–1750 mg/L. There was no observable growth for any population at 2500 mg/L (data not shown). 

### 3.2. Metal Resistance

Measurement of 24-h growth in excess metal (iron (II), iron (III), gallium (III), and silver (I)) was accessed for the FeNP, control, and ancestral populations at 25 days of evolution (Figure 2A–D). The FeNP showed highly statistically greater 24-h growth across concentration compared to the controls, which were greater than the ancestors in magnetite, iron (II), iron (III), gallium (III), and silver (I) (Table 2a). Iron (II) and iron (III) are of particular interest to this study because in solution, magnetite nanoparticles release both iron (II) and iron (III) ions into the medium. Measurements by ICP-OES determined how much of each of these species was released relative to the measured mass of that species added to DMB medium (Appendix A
Table A2, Figure A1, Figure A2 and Figure A3). These figures show that the actual amount of iron entering solution was always considerably less than the measured amounts initially added to the medium. The iron contents in these solutions were in the following order magnetite NPs > iron (III) > iron (II). The method presented was assessed by utilizing aliquots and dilutions of magnetite NPs, iron (II), and iron (III). The emission line used for the quantification of Fe (II) was 259.9 nm. Results showed high precision with the relative standard deviation (RSD) values for most of the samples < 2%. The plotted calibration curve was linear with a R^2^ value > 0.999.

The controls also showed statistically greater growth in all metals across concentration compared to the ancestor.

### 3.3. Antibiotic Resistance

The FeNP populations, controls, and ancestors were evaluated at 25 days of evolution for general antibiotic resistance (Figure 3A–D). FeNP populations showed superior growth in ampicillin and rifampicin compared to controls and the ancestral population at all concentrations (Figure 3A–D). FeNP populations showed significantly greater growth compared to the controls and ancestors for variable ranges in chloramphenicol (6–12 mg/L) and tetracycline (75–250 mg/L). The controls showed superior growth compared to the ancestral population in chloramphenicol (6–50 mg/L) and tetracycline 6–175 mg/L). All populations showed a reduction in growth with increasing concentration of antibiotics. All comparisons between FeNP and controls in antibiotics showed a significant interaction effect, indicating that the functional response to the antibiotic differed between these populations (Table 2a). Similarly, in the comparison of the controls to the ancestors, significant interaction effects were also seen for all antibiotics (Table 2b).

### 3.4. Cell Length and width Distribution

The size distribution of FeNP, control, and ancestral populations after 25 days of selection was determined by scanning electron microscopy (SEM). FeNP populations showed a significant increased cell length (2159.4 +/− 15.3 versus 1310.4 +/− 24.1 nm) and width (582.0 +/− 1.63 versus 471.8 +/− 16.2 nm) compared to the control and ancestral populations (F = 395.9, *p* < 0.000; Figure 4). There was no replicate effect for length (F = 0.255, *p* < 0.906); but there was significant population by replication effect (F = 4.32, *p* < 0.002). The controls were also slightly longer in length compared to the ancestral population 1310.4 +/− 24.1 versus 1248 +/−14.4 nm; F = 4.5, *p* < 0.034.). There was a significant replicate effect and population by replicate interaction (F = 6.7, *p* < 0.001 and F = 2.5, *p* < 0.041 respectively). The population effect on cell width was F = 79.4, *p* < 0.001; replicate effect was F = 8.6, *p* < 0.001; with an interaction for population and replicate at F = 8.6, *p* < 0.001 for the comparison of the FeNP with the controls. The mean cell width for the ancestors was not significantly different from the controls (F = 0.493, *p* < 0.483). There was significant variation within the ancestral population replicates for cell width (F = 8.1, *p* < 0.001) as well as significant interaction between population and replicates (F = 6.2, *p* < 0.001).

### 3.5. Whole Genome

The genomic variants found in the FeNP are listed in Table 3a–c. Descriptions of these genes are given in Table 4a–c. The genomic variants for the controls are provided in Appendix B
Table A3 and Table A4. The controls showed no hard selective sweeps. Furthermore, genomic variants for the ancestral population are not reported here as these are given in our prior research [41,42]. The ancestral variants were filtered out of all variant calls in the FeNP and Control populations. At 25 days, three of the five FeNP replicates (FeNP_1_, FeNP_2_, and FeNP_5_) displayed a hard selective sweep for non-synonymous (NS) substitutions in the RNA polymerase subunit β’ (*rpoC*) gene. Significant polymorphisms in RNA polymerase subunit α (*rpoA*) were observed in FeNP_3_ and FeNP_5_ (f = 0.503 and 0.665, respectively). FeNP_4_ was the only population displaying no hard selective sweeps but had significant polymorphisms in the flagellar hook protein (*flgE*) and NADH: ubiquinone oxidoreductase, chain G (*nuoG*) genes (f = 0.295, 0.439 respectively).

Significant polymorphisms (f > 0.250) were observed in all replicates (Table 3b). FeNP_1_ showed NS polymorphisms in porphyrinogen oxidase, cytoplasmic (*yfeX*), and putative metal-dependent hydrolase (*ygjP*). FeNP_2_ had NS polymorphisms in hydroxyacylgluthione hydrolase (*gloB*) and a synonymous substitution in the glutamate/aspartate: proton symporter (*gltP*). Synonymous substitutions may result from linkage to a NS beneficial mutation, or they can result from favorable selection due to codon bias. FeNP_3_ had NS polymorphisms in RNA polymerase α subunit (*rpoA*) and a synonymous variant in chromate reductase, Class I, flavoprotein (*chrR*). FeNP_4_ in flagellar hook protein (*flgE*) and NADH ubiquinone oxidoreductase, chain G (*nuoG*). Finally, FeNP_5_ also showed a significant NS polymorphism in RNA polymerase, α subunit (*rpoA*). Minor polymorphisms are listed in Table 3c. The controls displayed significant NS polymorphisms in RNA polymerase subunits β and β’. These are often associated with adaptation to minimal medium such as DMB. The *rpoB* H526Y variant has been repeatability observed arising in our control populations in several of our past studies [41,42].

## 4. Discussion

The study examined the potential for *E. coli* K-12 MG1655 to evolve magnetite nanoparticle resistance. It also examined the genomics changes associated with magnetite nanoparticle resistance. We demonstrated that by day 25, increased magnetite NP resistance was apparent in populations cultured in magnetite NPs (FeNP_1–5_). The FeNP replicates displayed highly statistically different increases (between 50 and 2 times greater) 24-h growth across concentration in magnetite with a mean increase across concentration of 8.5 times. They were highly statistically significantly higher in their capacity to grow in magnetite compared to the controls by similar margins. The controls display adaptations to growth in DMB medium that are not seen in the ancestors. As in our previous experiments, the controls performed better than the ancestors in the stress related assays (magnetite, ionic metals, and antibiotics). This is related to their overall improvement in growth associated with adaptation to DMB medium. As far as we know, this is the first report of the experimental evolution of magnetite resistance in the literature.

Resistance to magnetite NPs also conferred greater fitness in increasing concentrations to ionic metals iron (II), iron (III), gallium (III), and silver. The FeNP populations showed highly statistically significantly greater resistance to all of these metals compared to controls and ancestors as well. Similarly, FeNP populations showed significantly greater resistance to ampicillin, chloramphenicol, rifampicin, sulfanilamide, and tetracycline compared to controls and ancestors. 

One of the most striking results of this study was the shift in cell proportions in the FeNP populations compared to the controls and ancestors. The FeNP populations increased in both their length and width compared to both. As *E. coli* is a rod-shaped bacterium, a rough calculation of the increase in the cell area is given by multiplying its length and width. On average FeNP area = 1.25 × 10^6^ nm^2^, control = 6.17 × 10^5^ nm^2^, and ancestor = 5.76 × 10^5^ nm^2^. This represents a 2.18-fold increase in cell area in the FeNP populations compared to their ancestor. The magnitude of the cell size response suggests that this may be playing an important role in FeNP adaptation to metal and antibiotic toxicity. One of the primary mechanisms by which magnetite is known to impact bacteria is via ROS [45]. There is evidence that larger cell size is correlated to ROS resistance. A study of *Lactobacillus* species found that greater cell size was positively associated with resistance to ROS [46]. Another study of *Mycobacterium* isolates found that greater size variation was associated with antibiotic resistance [47]. In *E. coli* MG1655, stress is known to cause an increased cell length; this has also been observed in *M. tuberculosis* and *M. smegmatis* [48,49]. Finally, it has been shown that increase in cell length is an adaptation associated with antibiotic resistance due to the synthesis of a modified nucleotide in response to stress (p) ppGpp [50,51,52,53].

Genetics analysis identified selective sweeps in RNA polymerase subunit β’ (*rpoC*) in some replicates. In addition, others displayed significant polymorphisms (f > 0.500) in RNA polymerase subunit α (*rpoA*). As RNA polymerase is involved in putting together all the RNA in the cell, mutations in this protein are known to be widely pleiotropic and play major roles in relieving stress and increasing metabolic efficiency in *E. coli* [54,55]. This may explain why the FeNP populations displayed superior growth compared to the control and ancestral populations in all the stresses applied to them in this study (magnetite, ionic metals, and antibiotics). It can be argued that selective sweeps displayed in specific genes associated with a given environment at best illustrate an indirect demonstration of the causal role of those genes in producing resistant phenotypes. This is a general limitation of experimental evolution as a means of studying microbial adaptation [14,41]. However, in prior studies we have deployed more direct methods to test whether the genomic variants we discovered should and actually do confer resistance. For example, we have utilized computational modeling of 3-dimensional structures of altered proteins to test their affinity for ionic metals [12,15,42]. These studies showed the alteration in the protein caused by the variant resulting from the selective sweep did reduce or increase the affinity to the target metal compared to the controls/ancestor. We also have deployed recombination to move genetic variants discovered by experimental evolution into the genomic background of the ancestor to demonstrate that the variant did produce the resistance observed in the selected populations [42]. We did not perform these experiments in this study, simply due to the abundance of evidence that experimental evolution does reliably uncover adaptive variants.

Despite the fact that the FeNP populations displayed an increase in growth in magnetite, iron (II), iron (III), and gallium (III, an iron III analog), we did not find selective sweeps of variants associated with iron resistance in our prior studies [14,15,16]. These included mutations in genes such *as fecA*, *fur*, *dnaK*, *murC*, *ptsP*, and *ilvG*. Magnetite contains both iron (II) and iron (III), and these are released in solution [1]. In this study, the γ-magnetite particles were stabilized by a coating of tetramethylammonium hydroxide (CH_3_)_4_NOH. In our study, the magnetite nanoparticles were stabilized with PVP (polyvinylpyrrolidone). Coating agents used to improve the stability of magnetite nanoparticles decrease the release of ionic Fe^2+^ and Fe^3+^ in the medium [56,57]. 

Resistance to magnetite NP conferred resistance to all the antibiotics tested similarly to the iron (II)- and iron (III)-resistant populations from our previous studies [13,14]. These antibiotics have different modes of action due to the nature of their structures and degree of affinity to certain target sites within bacterial cells. Ampicillin exerts bactericidal activity through inhibition of bacterial cell wall synthesis by binding to penicillin binding proteins (PBPs) and by inhibiting certain PBPs related to the activation of a bacterial autolytic process [58]. Tetracycline enters the cells and binds reversibly to the 30S arresting translation, thus inhibiting protein synthesis which ultimately leads to a bacteriostatic effect [59]. Sulfanilamide prevents bacterial replication by inhibiting dihydropteroate synthetase [60]. Chloramphenicol diffuses through the bacteria cell wall and reversibly bind to bacterial 50S ribosomal subunit interfering peptidyl transferase activity and block peptide bond formation impeding bacterial cell proliferation [61]. Rifampin specifically inhibits bacterial RNA polymerase arresting DNA-directed RNA synthesis of bacteria [62]. Resistance to these antibiotics may involve efflux systems that transport the antibiotics from inside to outside the bacterial cells or a ribosomal protection protein remove antibiotics from ribosomes. Resistance to rifampicin has been linked to a variety of mutations in *rpoB* and *rpoC* [63,64,65]. Given that the FeNP populations have a general lack of variants in genes traditionally associated with iron (II), iron (III), or antibiotic resistance, we propose that this resistance may result from changes in gene expression. 

Finally, while we did not measure changes in gene expression in this study, our past study of iron (II) resistance in *E. coli* K-12 MG1655 found that there were large changes in gene expression associated with general metal resistance and iron metabolism in general [19]. We suspect that this was the case here as well and our future studies will examine this possibility. 

## 5. Conclusions

This is the first study to utilize experimental evolution to investigate magnetite nanoparticle resistance in *E. coli* or any other bacterium. *E. coli* K-12 MG1655 evolved resistance to magnetite nanoparticles at 25 days of selection. Resistance to magnetite nanoparticles conferred resistance to both metals and antibiotics. Increased cell length and wide resulted many have played a significant role in adaptation to magnetite. Genomic analysis revealed hard selective sweeps in the *rpoA* and *rpoC* made important contributions to magnetite nanoparticle resistance. These results have important consequences for the future use of magnetite nanoparticles as antimicrobials as de novo evolution against these materials resulted quickly and occurred due to relatively simple genomic changes. This is an issue that will continue to be of significance regarding the use of nanomaterials as antimicrobials as we outlined in an earlier review [66].

## Figures and Tables

**Figure 1 nanomaterials-11-00790-f001:**
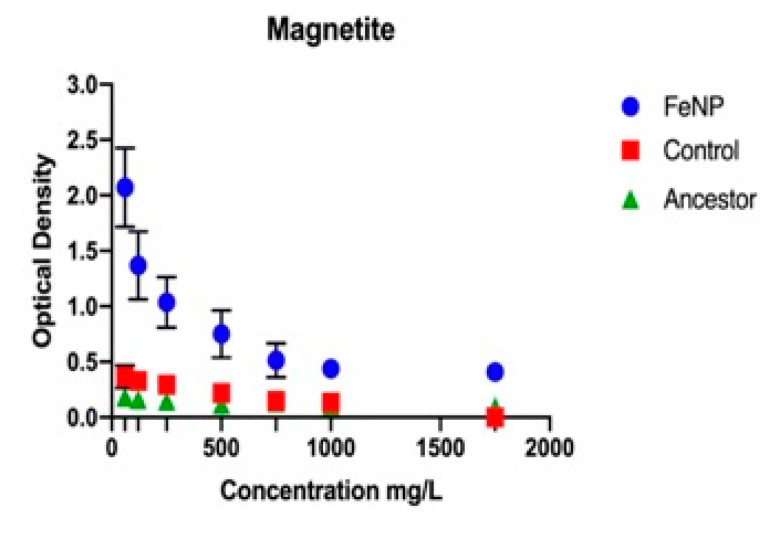
The mean and SE of 24 h-hour growth for FeNP, Control, and Ancestral populations in increasing concentration of Magnetite NPs are shown. There was no observable growth at 2000 mg/mL magnetite for the FeNP population.

**Figure 2 nanomaterials-11-00790-f002:**
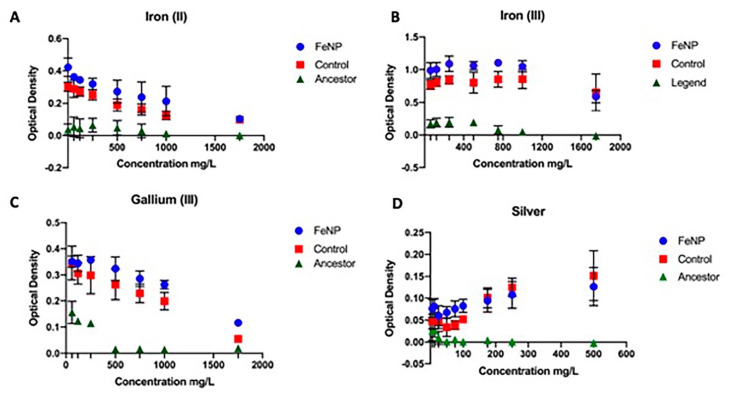
The mean and SE of 24h-hour growth for FeNP, Control, and Ancestral populations in increasing concentration of metals are shown. The FeNP populations showed superior growth compared to controls for iron (II, III, **A**,**B**) and gallium (III, **C**) from 6–1750 mg/L. For silver superiority was shown from 6–100 mg/L (**D**). The ancestor showed inferior growth to the FeNPs and controls at all concentrations.

**Figure 3 nanomaterials-11-00790-f003:**
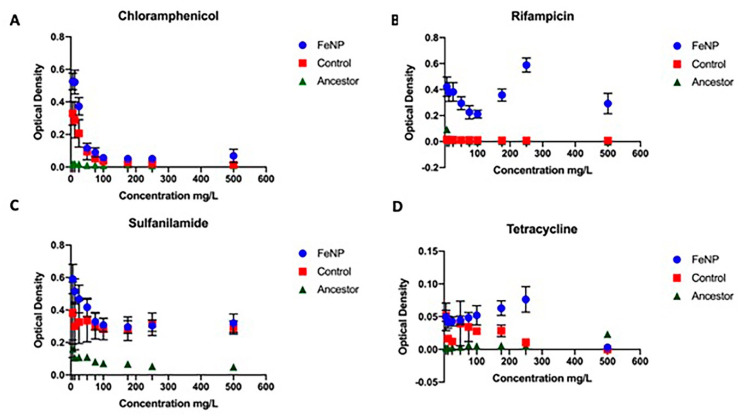
The mean and SE of 24h-hour growth for FeNP, Control, and Ancestral populations in increasing concentration of antibiotics are shown. The FeNP populations showed superior growth compared to controls and ancestors for chloramphenicol at 25, 60 mg/L (**A**); rifampicin (**B**) from 25–500 mg/l; sulfanilamide (**C**) from 50–75 mg/L; and tetracycline (**D**) from 75–250 mg/L. The ancestor showed inferior growth to the FeNPs and controls at all concentrations.

**Figure 4 nanomaterials-11-00790-f004:**
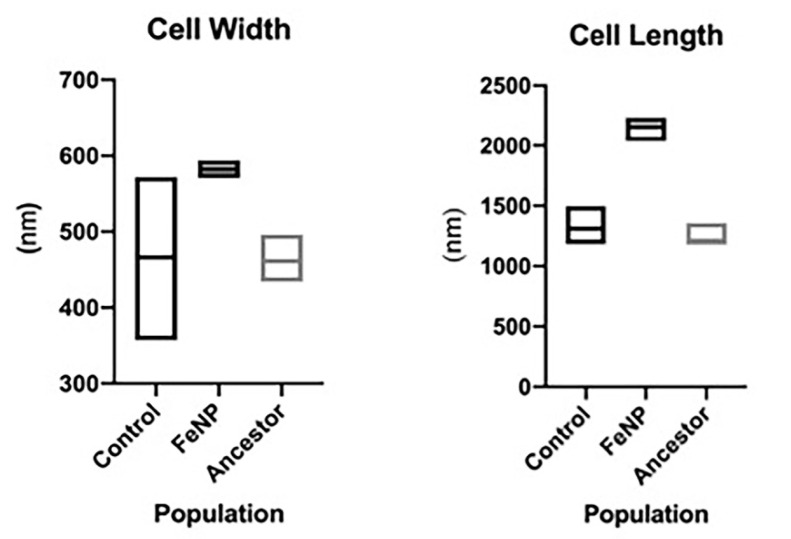
**Bacteria cell width and length distribution**. Box plots of *E. coli* MG1655 mean cell width and length for FeNP compared to controls and ancestors is shown. The sample size was N = 20 cells per replicate population measures in the FeNP and control populations for a total of 100 cells per selection treatment. Sample size for ancestors was 15 for each replicate, for a total of 75 cells.

**Table 1 nanomaterials-11-00790-t001:** Mechanisms of excess iron and silver toxicity: Mechanisms of cellular damage are listed resulting from excess iron toxicity. As these systems are common to virtually all bacteria, there is a strong potential that resistance mechanisms might be conserved across wide varieties of taxa.

Mechanism	Fe
Reactive oxygen species	+
Disruption of transcription/translation	+
Damage to cell wall/membrane	+
Interfering with respiration	+
Release of cellular components	+
Binding to thiol groups	+

**Table 2 nanomaterials-11-00790-t002:** (**a**) General linear model results for phenotypic assays, FeNP vs. Control. (**b**) General linear model results for phenotypic assays, control vs. ancestor.

(a)			
Substance	Pop. Effect (F, p)	Conc. Effect (F, p)	Interaction (F, p)
magnetite	383.9, 0.001	42.5, 0.001	25.9, 0.001
ampicillin	283.1, 0.001	17.1, 0.001	16.7, 0.001
chloramphenicol	71.1, 0.001	114.6, 0.001	7.9, 0.001
rifampicin	1285.1, 0.001	16.3, 0.001	16.5, 0.001
tetracycline	62.9, 0.001	12.6, 0.001	5.7, 0.001
iron (II)	18.6, 0.001	88.0, 0.001	1.7, 0.116
iron (III)	42.9, 0.001	77.4, 0.001	2.7, 0.011
gallium (III)	29.8, 0.001	111.8, 0.001	1.2, 0.278
silver (I)	7.1, 0.009	15.8, 0.001	2.3, 0.029
**(b)**			
**Substance**	**Pop. Effect (F, p)**	**Conc. Effect (F, p)**	**Interaction (F, p)**
magnetite	9.5, 0.003	22.7, 0.001	12.8, 0.001
chloramphenicol	159.1, 0.001	25.7, 0.001	22.6, 0.001
silver (I)	243.1, 0.001	10.4, 0.001	15.6, 0.001

**Table 3 nanomaterials-11-00790-t003:** (**a**) Selective sweeps in FeNP-resistant populations at day 25. (**b**) Significant polymorphisms in FeNP-resistant populations at day 25. (**c**) Minor polymorphisms in FeNP-resistant populations at day 25.

(a)							
Gene	Mutation	Annotation	FeNp1	FeNp2	FeNp3	FeNp4	FeNp5
rpoC→	A→T	D410V (GAT→GTT)	0.000	0.000	0.000	0.000	1.000
rpoC→	A→G	D622G (GAC→GGC)	1.000	1.000	0.000	0.000	0.000
**(b)**							
**Gene**	**Mutation**	**Annotation**	**FeNp1**	**FeNp2**	**FeNp3**	**FeNp4**	**FeNp5**
gloB←	T→A	E239V (GAG→GTG)	0.000	0.277	0.000	0.000	0.000
flgE→	C→T	A28A (GCC→GCT)	0.000	0.000	0.000	0.295	0.000
nuoG←	C→T	G792S (GGT→AGT)	0.000	0.000	0.000	0.439	0.000
yfeX←	G→T	L263M (CTG→ATG)	0.251	0.000	0.000	0.000	0.000
ygjP→	A→G	Q17R (CAG→CGG)	0.270	0.000	0.000	0.000	0.000
rng←	T→A	H76L (CAC→CTC)	0.000	0.000	0.000	0.250	0.000
rpoA←	C→A	D199Y (GAC→TAC)	0.000	0.000	0.503	0.000	0.000
rpoA←	G→A	R191C (CGT→TGT)	0.000	0.000	0.000	0.000	0.665
chrR→	G→A	P32P (CCG→CCA)	0.000	0.000	0.258	0.000	0.000
gltP→	A→G	G250G (GGA→GGG)	0.000	0.302	0.000	0.000	0.000
**(c)**							
**Gene**	**Mutation**	**Annotation**	**FeNp1**	**FeNp2**	**FeNp3**	**FeNp4**	**FeNp5**
yagL←	T→A	E93V (GAG→GTG)	0.000	0.165	0.000	0.000	0.000
gapC←	C→T	pseudogene (254/750 nt)	0.000	0.000	0.134	0.000	0.000
lsrD→	A→T	N258I (AAT→ATT)	0.000	0.000	0.159	0.000	0.000
pta→	C→T	R669C (CGT→TGT)	0.000	0.207	0.000	0.000	0.000
intS→	G→A	G77S (GGC→AGC)	0.000	0.175	0.000	0.000	0.000
yraJ→	A→G	K85E (AAG→GAG)	0.154	0.000	0.000	0.000	0.000
prmA/dusB	A→T	intergenic (+317/-12)	0.000	0.000	0.000	0.000	0.209
hdeB←	A→T	N31K (AAT→AAA)	0.000	0.000	0.000	0.000	0.155
viaA←	T→A	E153V (GAA→GTA)	0.000	0.000	0.161	0.000	0.000
frwA←	C→T	E363E (GAG→GAA)	0.000	0.000	0.183	0.000	0.000
treR←	C→T	P275P (CCG→CCA)	0.000	0.000	0.000	0.000	0.111
sgcX←	C→A	Q155H (CAG→CAT)	0.152	0.000	0.000	0.000	0.000

**Table 4 nanomaterials-11-00790-t004:** (**a**) Position and description of selective sweeps in FeNP-resistant populations at day 25. (**b**) Position and description of significant polymorphisms in FeNP-resistant populations at day 28. (**c**) Position and description of minor polymorphisms in FeNP-resistant populations at day 25.

(a)		
Gene	Position	Description
rpoC→	4,186,578 & 4,187,214	RNA polymerase, beta prime subunit
**(b)**		
**Gene**	**Position**	**Description**
gloB←	234,067	hydroxyacylglutathione hydrolase
flgE→	1,132,657	flagellar hook protein
nuoG←	2,397,792	NADH: ubiquinone oxidoreductase, chain G
yfeX←	2,549,759	porphyrinogen oxidase, cytoplasmic
ygjP→	3,236,009	putative metal dependent hydrolase
rng←	3,397,569	ribonuclease G
rpoA←	3,440,435 & 3,440,459	RNA polymerase, alpha subunit
chrR→	3,894,747	chromate reductase, Class I, flavoprotein
gltP→	4,295,230	glutamate/aspartate: proton symporter
**(c)**		
**Gene**	**Position**	**Description**
yagL←	293,641	CP4-6 prophage; DNA-binding protein
gapC←	1,490,460	pseudogene, GAP dehydrogenase; 1.
lsrD→	1,604,819	autoinducer 2 import system permease protein
pta→	2,416,751	phosphate acetyltransferase
intS→	2,466,773	CPS-53 (KpLE1) prophage; putative prophage CPS-53 integrase
yraJ→	3,289,066	putative outer membrane protein
prmA/dusB	3,410,268	methyltransferase for 50S ribosomal subunit protein L11/ 2.
hdeB←	3,656,200	acid-resistance protein
viaA←	3,929,146	stimulator of RavA ATPase activity; 3.
frwA←	4,141,133	putative PTS enzyme, Hpr component/enzyme I component/ 4.
treR←	4,466,422	trehalose 6-phosphate-inducible trehalose regulon 5.
sgcX←	4,531,187	putative endoglucanase with Zn-dependent exopeptidase domain

1. glyceraldehyde-3-phosphate dehydrogenase (second fragment). 2. tRNA-dihydrouridine synthase B 3. von Willebrand factor domain protein 4. enzyme IIA component 5. transcriptional repressor.

## Data Availability

The SRA accession number for sequencing data is PRJNA694183 (FeNP-resistant, FeNP_1–5__D28 and controls, C_1–5__D28). Finally, the phenotypic data from these studies will be submitted into DRYAD (https://datadryad.org/ accessed on 17 March 2021) upon acceptance of this manuscript for publication.

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
