# Peer review of "Experimental Evolution of Magnetite Nanoparticle Resistance in Escherichia coli"

_nanomaterials, 2021, doi:10.3390/nano11030790_

Round 1

Reviewer 1 Report

This is a nicely designed experimental evolution study, to examine phenotypic and molecular evolution of E. coli bacteria in response to selection pressure imposed by magnetite. The authors employed useful controls, that compare treatment outcomes to bacterial evolution for a similar time period in the absence of magnetite selection. Overall, the methods and analyses were appropriate, and the conclusions are reasonable. However, I have some concerns regarding the study, listed below:

Major concerns

  1. The evolution experiment utilized magnetite as a selective agent, but the authors tested for evolved changes in bacterial fitness in many other environments, such as utilizing various antibiotics. By the time I finished reading the abstract and introduction, I was left wondering why these other environments were examined, because there is no obvious explanation offered. In particular, the authors could have included these assays based on some knowledge (in the literature or in their direct experience) of mechanistic reasons to expect selection in one environment to alter performance (correlated improvement vs. fitness trade-off) in the others. The text that closest addresses this omission does not appear until the Discussion, and it still fails to provide a sound explanation. The paragraph in the middle of page 12 nicely describes the similarities and differences among the various antibiotics and their actions, but this text reads like a review on antibiotics themselves, where there is no attempt to link the information to WHY the antibiotic assays were included here. The authors should try harder to justify why they included these other phenotypic traits, either for mechanistic reasons or simply due to motivations of studying various antibacterials side by side because of their applied significance (e.g., if one selective agent is used, should we worry about evolutionary change in the presence of other selective agents?). Thus, they could use the opportunity to describe correlated responses to selection versus trade-offs in performance, as a way to enrich the evolutionary biology presented in the study. I believe this is worthwhile (and perhaps necessary) to be very clear on these subjects, since the core audience of the journal is not researchers trained in evolutionary biology. In short, without some better context, the inclusion of the various other challenge assays seems rather arbitrary, which distracts from the other strengths of the study.
  2. I am confused why the controls improved growth in all metals compared to the ancestor (page 6). Is this simply due to generalized selection that alters bacterial metabolism?
  3. Some inferred genetic evidence for the phenotypic changes is presented utilizing the genome sequencing results, and this is discussed (page 12). But there was no recombinant work done to try and narrow down the genetic basis for the observed changes in characters. I gauge that the authors believe this is beyond the scope of the present study. Nevertheless, the discussion of these results on lines 357-364 is rather weak. The authors should expand this paragraph a bit, otherwise the genetics data are very to-the-side, and do not add much richness to the paper.
  4. Throughout the manuscript there are grammar errors, which presumably could have been fixed before submission by one of the many co-authors in the event that the main writer(s) did not catch them. Ordinarily, I would state this as a minor concern, but I chose to highlight it here because there are many grammar fixes needed. For example, on page 4, line 174 it should state: “Reads that show a distinction …”.

Minor concerns

  1. Page 2, line 48. I believe there is a typo in this sentence, and it should read “stressors” instead of “stress”.

Author Response

In response to the reviewer's concern about why other environments were examined, I have added text on Line 67 explaining that our past studies of excess iron (II) and iron (III) found unanticipated correlated responses to antibiotics.

In response to the reviewer's question about superior growth of controls over ancestors I added text on lines 335-338 explained general adaptation of controls to DMB medium resulting in their superior performance as observed in all of our past studies.

In response to reviewer's concern about indirect evidence of adaptation I added text to lines 357-364 explaining how we provided direct evidence that experimental evolution uncovered causal variants in our previous studies via our computational modeling, and recombination of variants into ancestral genomic backgrounds.  We did not perform those studies here, but plan to do that work in the studies that will examine gene expression.

In response to the reviewer's concern about grammar, I reran grammar/spell check and visually inspected manuscript to find any missed grammatical errors.

Reviewer 2 Report

The main concept of the article is interesting and worth investigating. In general, the experiments performed have been properly designed but some improvements are strongly recommended to make manuscript more understandable and consistent. The comments and suggestions are as follows:

  • Dot should be removed from the title of the manuscript.
  • The background of the research as well as its novelty should be briefly indicated in Abstract.
  • First sentence of the Introduction needs to be corrected. Ionic or nanoparticle iron can not be defined as methods but as agents (or materials) used to control the multidrug resistant bacteria. Additionally, such wide range of references as [26-31] is not appropriate. In such a situation, Authors should divide this citation into few smaller ones and discuss briefly the information from particular literature reports important in viewpoint of the topic undertaken in the Introduction of the article.
  • Section 2.2.: what was the form of nanoparticles obtained, i.e. a suspension (concentration?) or a powder?
  • Data presented in Table 2a/2b are poorly understandable and given in a slightly unclear manner. Therefore, the discussion over them should be supplemented with a specific example with the description of all parameters presented in Table 2a/2b for a specific substance (using specific values of Pop. Effect, Conc. Effect and Interaction; e.g. what does mean for ampicillin Pop. Effect 283.1 and Interaction 16.7?). Now it is difficult to find a connection between the data presented in Tables and the discussion above them.
  • All figures in the article should be defined as “Figure” because now in some cases there is only a description without the name “Figure” and the adequate number, e.g. figures on pages 7-9. Caption of Figure 1.: it should indicate what is shown in the figure without any conclusions.
  • The abbreviation FeNPs is sometimes defined as magnetite nanoparticles and sometimes it refers to the population resistant to magnetite nanoparticles. It needs to be corrected because the abbreviation used should be clear and unambiguous.
  • From the editorial point of view, Authors should pay more attention to the consistency in the notation of units, i.e. in some cases the notation mg/L is applied and in other cases – mg/l etc. or sometimes the notation 37°C is applied and sometimes 37 °C (with space between the number and the unit).
  • Section References should be improved to be consistent. Now some literature reports contain the whole journal names and the other ones contain their abbreviations.

Author Response

I have responded to all of this reviewers concerns: background and novelty in abstract, explained general linear models more fully, found and corrected all stylistic concerns and grammatical errors.